# Quantitative proteomics of infected macrophages reveals novel *Leishmania* virulence factors

Nicolas Hagedorn[1☉], Albert Fradera-Sola[2☉], Melina Mitnacht[1], Tobias Gold[3], Ulrike Schleicher[3], Falk Butter[2]*, Christian J. Janzen[1]*

**1** Department of Cell & Developmental Biology, Biocentre, University of Würzburg, Würzburg, Germany, **2** Institute of Molecular Virology and Cell Biology, Friedrich-Loeffler-Institute, Greifswald, Germany, **3** Institut für Klinische Mikrobiologie, Immunologie und Hygiene, Universitätsklinikum Erlangen, Friedrich-Alexander Universität Erlangen-Nürnberg, Erlangen, Germany

☉ These authors are contributed equally.
\* christian.janzen@uni-wuerzburg.de (CJJ); falk.butter@fli.de (FB)

## Abstract

Leishmaniasis is a major public health problem, causing diseases ranging from self-healing skin lesions to life-threatening chronic infections. Understanding how *Leishmania* parasites evade the host defense system is crucial for understanding the different manifestations of the disease and for improving diagnostic tools and drug development. We performed high-resolution proteome profiling of *Leishmania* spp. across three species during macrophage infection and identified distinct temporal expression patterns. Clustering analysis revealed unique protein expression profiles for each *Leishmania* species, whereas pairwise enrichment analysis revealed specific up- and downregulation patterns at different infection stages. Our results confirmed known virulence factors and highlighted new ones, demonstrating how our dataset could be used. We validated the dataset by showing that deletion of putative *L. mexicana* virulence factors resulted in reduced stage differentiation capacity and infectivity.

## Summary

Leishmaniasis is a devastating neglected tropical disease caused by the obligate intracellular protozoan *Leishmania,* with an estimated 700.000 to 1 million new cases annually worldwide. There are three main manifestations of leishmaniases: the most common cutaneous form, the mucocutaneous form and the most severe visceral form, which is almost always fatal without treatment. Despite decades of fruitful research, many important questions remain unanswered. For example, how can different *Leishmania* species cause different manifestations of the disease? In fact, *Leishmania infantum* can cause both, visceral leishmaniasis, which is relatively common, and cutaneous leishmaniasis. Furthermore, *Leishmania* parasites have developed a sophisticated life cycle, including an intracellular stage in phagocytic cells of their vertebrate hosts, such

---

**Data availability statement:** Data are available via ProteomeXchange with identifier PXD058427.

**Funding:** The author(s) received no specific funding for this work.

**Competing interests:** The authors have declared that no competing interests exist.

as macrophages. However, how parasites evade the host immune response to survive and proliferate in this hostile environment is still elusive. Quantitative mass spectrometry is a powerful tool for analyzing how parasite proteomes need to change to adapt to challenges of the host defense system to establish a chronic infection. We therefore conducted this proteomics study with three different *Leishmania* species during macrophage infection to generate a resource platform that provides additional data to ongoing studies in the community or serves as starting points for new projects.

## Introduction

The protozoan parasites *Leishmania* spp. are the causative agents of the zoonotic disease leishmaniasis. It is considered a neglected tropical disease with an estimated 1 million infections annually and an increasing prevalence among those with limited access to healthcare and poor hygiene infrastructure [1]. The disease pathology can be classified into three groups with increasing severity: cutaneous (CL), mucocutaneous (MCL) and visceral (VL) leishmaniasis [2]. The course of pathology depends on multiple factors, including the parasite species and the host's immune response [3]. *L. mexicana* and *L. major* cause CL, where skin lesions or ulcers occur at the site of infection [4]. MCL is usually caused by *Leishmania* spp. of the subgenus *Viannia* (e.g., *L. braziliensis*) [4], but case reports have shown that MCL can also be caused by *L. major* [5–6]. MCL pathology is characterized by destruction of the mucosal tissue of the nose, mouth, and throat, and is potentially life-threatening [4]. VL can occur after infection with, for example, *L. infantum* and is characterized by enlargement of the liver and spleen and is, in most cases, lethal if left untreated [7]. To date, there is no available vaccine to prevent leishmaniasis in humans. The few available drugs are expensive and require long-term treatment [2–4], which is often not feasible in many endemic regions. Additionally, increasing drug resistance has been reported [8–9], increasing the difficulty of treating this disease.

Infection with *Leishmania* spp. Occurs primarily through the bite of female sand flies of the genera *Phlebotomus* and *Lutzomyia* [10–12]. They carry extracellular promastigote forms and transmit them to the mammalian host during a blood meal. Then, professional phagocytes, such as neutrophils, dendritic cells, and mononuclear macrophages, which are recruited to the site of infection, take up promastigotes via phagocytic pathways [7–13]. Once inside the host cell, the parasites interfere with phagocytic pathways, resulting in a delay of phagolysosome maturation and, thus, prevent the elimination of the parasites [14]. Virulence factors, such as lipophosphoglycan (LPG) or the metalloprotease GP63, are secreted by the parasite into the phagolysosome lumen, thus helping to transform the host compartment into the less hostile parasitophorous vacuole (PV). GP63 helps recruit early endosomal components such as Rab5a to the PV surface, which delays maturation [15]. LPG recruits protein kinase Cα (PKCα), which facilitates the degradation of F-actin around the PV, thus further delaying maturation [16–17]. The mechanisms used to establish a PV

are species specific and host cell dependent [18]. Among those differences are the morphologies of the PVs. *L. mexicana* resides and replicates in communal PVs, whereas *L. infantum* and *L. major* reside in individual PVs [18]. The PV provides an interface between the parasite and the host as well as an environment for the differentiation of the motile promastigote to the immotile amastigote form [19–21]. Differentiation is triggered by acidification of the PV as well as exposure to the higher temperature presented by the mammalian host. The *Leishmania* amastigotes are adapted to survive and proliferate in the harsh conditions they are exposed to in the PV. Here, virulence factors such as trypanothione reductase (TRYR) and trypanothione synthetase (TRYS) protect amastigotes against reactive oxygen species (ROS) generated by the host immune response [22–25]. Amastigotes are eventually released from the host cell and then taken up by other phagocytes [7]. The life cycle is completed when a sand fly takes a blood meal from an infected host.

The interaction between the parasite and the mammalian host cell is only partially understood. For example, the roles of LPG and GP63 in host evasion are well-studied, particularly their involvement in inhibiting phagosome maturation and suppressing ROS production [26–28]. On the other hand, the molecular details of the transition from promastigotes to amastigotes, as well as the mechanisms contributing to the distinct clinical pathologies of CL, MCL, and VL remain elusive. Few recent transcriptomics and proteomics studies have identified virulence factors but have focused on transcriptional and proteomic changes in either the host or the parasite [29–33], not both. Only one study has used a transcriptomics approach to compare the dynamic changes that occur during the process of infection with *L. amazonensis* and *L. major* parasites in human macrophages [34]. However, transcriptomic and proteomic data do not always correlate well in kinetoplastida [35] because the majority of the steady-state proteome is regulated by posttranscriptional mechanisms [36]. Thus, we designed a proteomic study that analyzes the dynamic changes in the protein composition of murine bone marrow-derived macrophages (BMDMs) and three *Leishmania* species upon infection. The extension of this study to three different *Leishmania* species allows us to identify important shared virulence factors that are present in multiple species and contribute to the pathogenicity of leishmaniasis, as well as species-specific virulence factors. Additionally, our study provides insight into possible similarities and differences between the parasites' proteome as well as the host response during infection with different strains. We also provide a comprehensive website that enables users to access and analyze the data for their specific scientific questions and applications.

## Results

### Proteome profiling of *Leishmania* spp. infection dynamics in murine macrophages

To characterize changes in the host cell and the parasite proteome, *M. musculus* bone marrow-derived macrophages (BMDMs) were infected with promastigotes of three *Leishmania* species with different clinical manifestations: *L. infantum*, *L. major*, and *L. mexicana*. We measured the parasite and the host cell proteomes in independently infected quadruplicates at seven time points post infection via label-free quantitative mass spectrometry (MS). Overall, we quantified 4,195 protein groups in *L. infantum*, 4,506 protein groups in *L. major* and 4,568 protein groups in the *L. mexicana* infection experiments (Fig 1A and S1 Table). All protein profiles can be interactively accessed via our *Leishmania* infectome database (https://butterlab.imb-mainz.de/LInfDB/).

For all three infection time courses, a similar number of ca. 3,000 murine protein groups were quantified (Fig 1A), with 59% shared protein group identifiers (Fig 1B). In contrast, the number of quantified *Leishmania* spp. proteins differed across the infection time courses: 1,075 for *L. infantum*, 1,407 for *L. major* and 1,523 for *L. mexicana* (Fig 1A). The lower number of *L. infantum* proteins correlates with its lower infection rates of macrophages and slightly lower number of parasites in individual BMDMs at late time points during the course of infection (S1 Fig). To compare the overlap of the quantified *Leishmania* spp. proteins, we determined the *Leishmania* spp. orthologs among the three species, proteome-wide, via OrthoMCL [37] and considered only one-to-one orthologs, accounting for approximately 93% of all annotated proteins (S2A Fig and S2 Table). This finding was comparable in our dataset, which matched 95.17% for *L. infantum*, 95.55% for

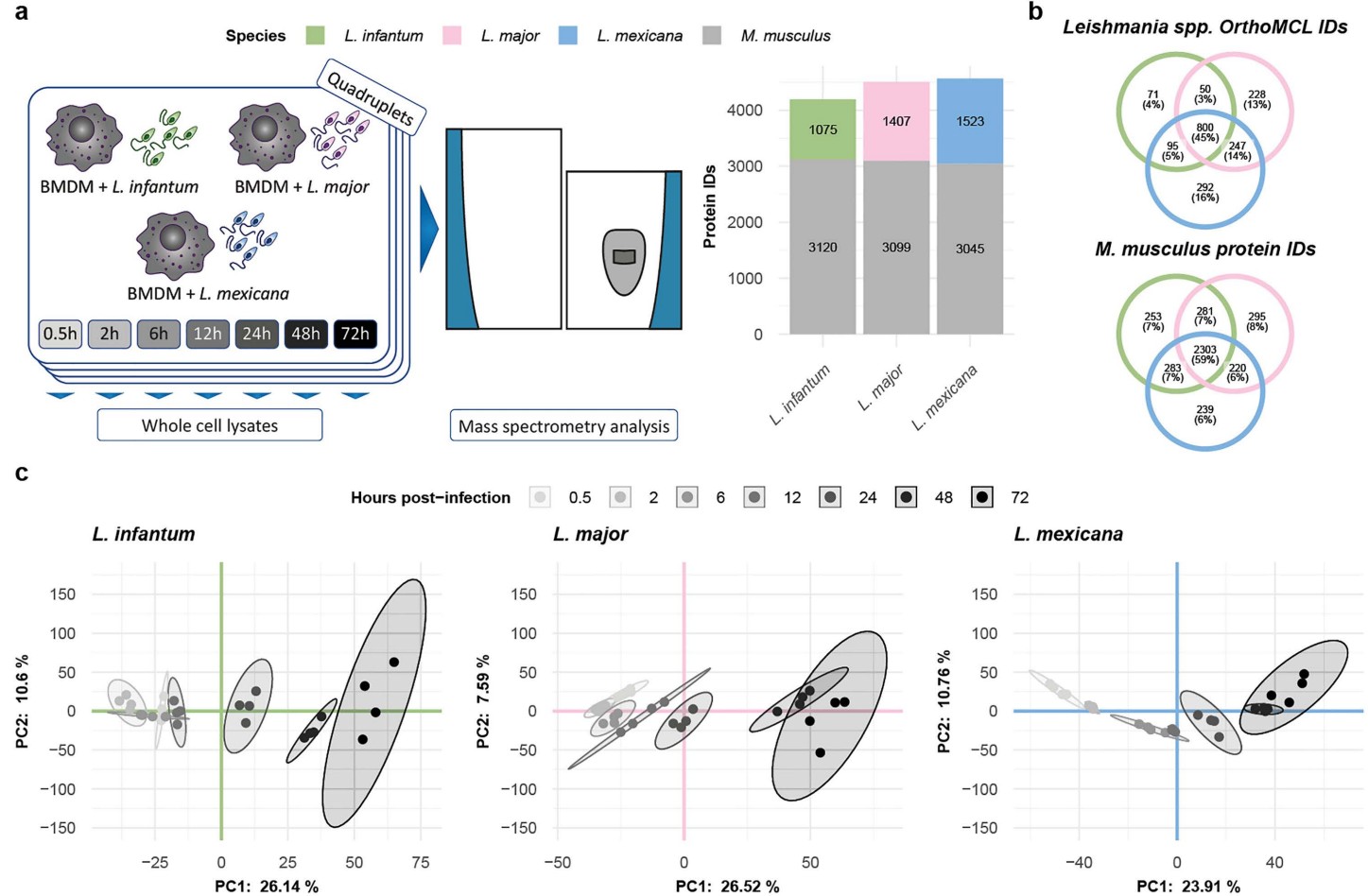

**Fig 1. Quantitative label-free mass spectrometry was used to quantify the *Leishmania* spp. and *M. musculus* proteomes at seven postinfection timepoints. a,** Three *Leishmania* spp. were used to infect *M. musculus* BMDMs in quadruplicate. Whole-cell lysates were prepared at seven different timepoints post infection and quantified via mass spectrometry. The bar plot represents the total number of *M. musculus* (gray) or *Leishmania* spp. (green, pink, and blue for *L. infantum*, *L. major*, and *L. mexicana*, respectively) quantified proteins in each experiment. **b,** Venn diagrams showing the overlap between the different *Leishmania* spp. OrthoMCL IDs and between the different *M. musculus* protein IDs obtained in the *L. infantum* (green), *L. major* (pink), and *L. mexicana* (blue) experiments. **c,** *Scatter* plot showing the first two components of the principal component analysis (PCA), which together explain 36.74%, 34.11%, and 34.67% of the variance in the data for the *L. infantum* (green), *L. major* (pink), and *L. mexicana* (blue) experiments, respectively. The amount of variance explained by each PC is indicated on each axis. The samples are represented as dots and color-coded according to a grayscale that increases in darkness on the basis of their postinfection timepoints. A 95% confidence level multivariate t distribution of each timepoint is represented as an ellipse, with colors corresponding to the grayscale used for sample representation.

*L. major*, and 95.05% for *L. mexicana* orthologs (S2B Fig). Not only did we quantify a lower number of proteins among the three species but also found few of them to be species-exclusive proteins: 4% for *L. infantum*, 13% for *L. major* and 16% for *L. mexicana* (Fig 1B). Notably, compared with the 59% overlap in murine protein groups, only 45% of *Leishmania* orthologs were quantified in all three species (Fig 1B).

To check for reproducibility, we performed a principal component analysis (PCA) on each infection time course and observed excellent agreement among the biological replicates (Figs 1C and S2B). For each time course, we observed a gradual transition between adjacent time points (Figs 1C and S2A). Additionally, we calculated the Euclidean distance between the first two PCA components of adjacent time points. This revealed differences in global protein expression

changes between time points for each infection time course. For example, compared with those of the other two species, the expression of the *L. mexicana* proteome changed more drastically between 2 and 6 hours post infection (hpi) (S3A Fig).

To further investigate these differences, we analyzed the protein expression dynamics of the individual proteins (Figs 2A and S4A). To this end, we calculated the Gini coefficient for protein abundance for each protein across the time courses and assigned stably and dynamically expressed proteins. This resulted in a set of 68 (*L. infantum*), 117 (*L. major*), and 195 (*L. mexicana*) proteins that exhibited highly dynamic expression changes (Gini score > 0.1 and $\log_2$(LFQ) > 27). (Figs 2B and S4B and S3 Table). In agreement with previous results in *Xenopus* and *Drosophila* [38–39], dynamicity was generally inversely correlated with protein abundance.

While mouse housekeeping proteins such as Myo9b were among the stably expressed proteins, known *Leishmania* spp. virulence factors and *M. musculus* inflammation markers were among the proteins with the greatest changes in abundance. For example, during infection with *L. infantum*, the protein levels of TRYR and peroxiredoxin 1M (PRX1M) first increased but subsequently decreased during infection (Fig 2C). The abundance of the *L. major* protein HSP20 increased with infection progression, whereas the abundance of TRYS decreased after 6 hpi. Interestingly, we observed an inverse expression pattern for TRYS in *L. mexicana*, with levels increasing until 6 hpi and then remaining constant. In the host, as expected, inflammation markers such as Toll-like receptor 2 (Tlr2) and interferon-induced protein with tetratricopeptide repeats 1 (Ifit1) increased during infection with any of the three *Leishmania* species.

## Exploring temporal protein dynamics with an artificial neural network

To generate groups of similarly expressed proteins, we clustered the data of the three infection time courses via self-organizing maps (SOMs). We performed separate analyses of the dynamically expressed proteins of the *Leishmania* spp. and the BMDM host cells. SOM analysis of the *Leishmania* spp. and *M. musculus* proteins revealed nine distinct clusters (Figs 3A, 4A, S5, and S6) each, with protein groups generally clustering according to their respective SOM cluster in the PCA (Figs 3B and 4B).

## Infection response of the host cell to different parasite species follows a similar trend

The murine protein groups quantified after infection with one of the three *Leishmania* spp. were generally equivalently represented among the clusters: none of the clusters had a time course specific protein fraction higher than 55% (Fig 3A and 3C and S4 Table). Notably, clusters 1 and 9 together captured 53% of the quantified murine proteins and showed opposite trends in their expression profiles: while protein abundance in Cluster 1 constantly decreased over the time course, in Cluster 9, it constantly increased. Thus, using the overlap of 219 protein groups in Cluster 1 and 137 protein groups in Cluster 9, we investigated the shared biological processes of each of the two clusters via Gene Ontology (GO) analysis. Functional enrichment analysis of Cluster 1 (S7A Fig and S5 Table) revealed processes such as DNA helicase activity (GO:0003678), chromatin DNA binding (GO:0031490), transcription factor binding (GO:0008134), and translation initiation factor activity (GO:0003743). Cluster 9 (S7A Fig and S5 Table) was enriched in GO terms such as oxidoreductase activity (GO:0016491) and proteoglycan binding (GO:0043394). We additionally queried for functional annotations within the HALLMARK database (S8 Fig and S6 Table). Among the results for Cluster 1, we found an overrepresentation for the GSE42088 UNINF VS LEISHMANIA INF DC 4H DN term; this term contains a collection of down-regulated genes obtained after 4h of infecting dendritic human cells with *L. major*.

Overall, these findings indicate that, regardless of the infecting *Leishmania* strain, there is a large overlap in *M. musculus* proteome expression changes, with a similar temporal pattern.

## Differential parasite protein expression during the infection process is species dependent

Six of the nine *Leishmania* SOM clusters were dominated by proteins from one of the species: Cluster 2 (61%) and Cluster 5 (78%) by *L. infantum*; Cluster 3 (72%) and Cluster 4 (84%) by *L. major;* and Cluster 8 (86%) and Cluster 9 (83%)

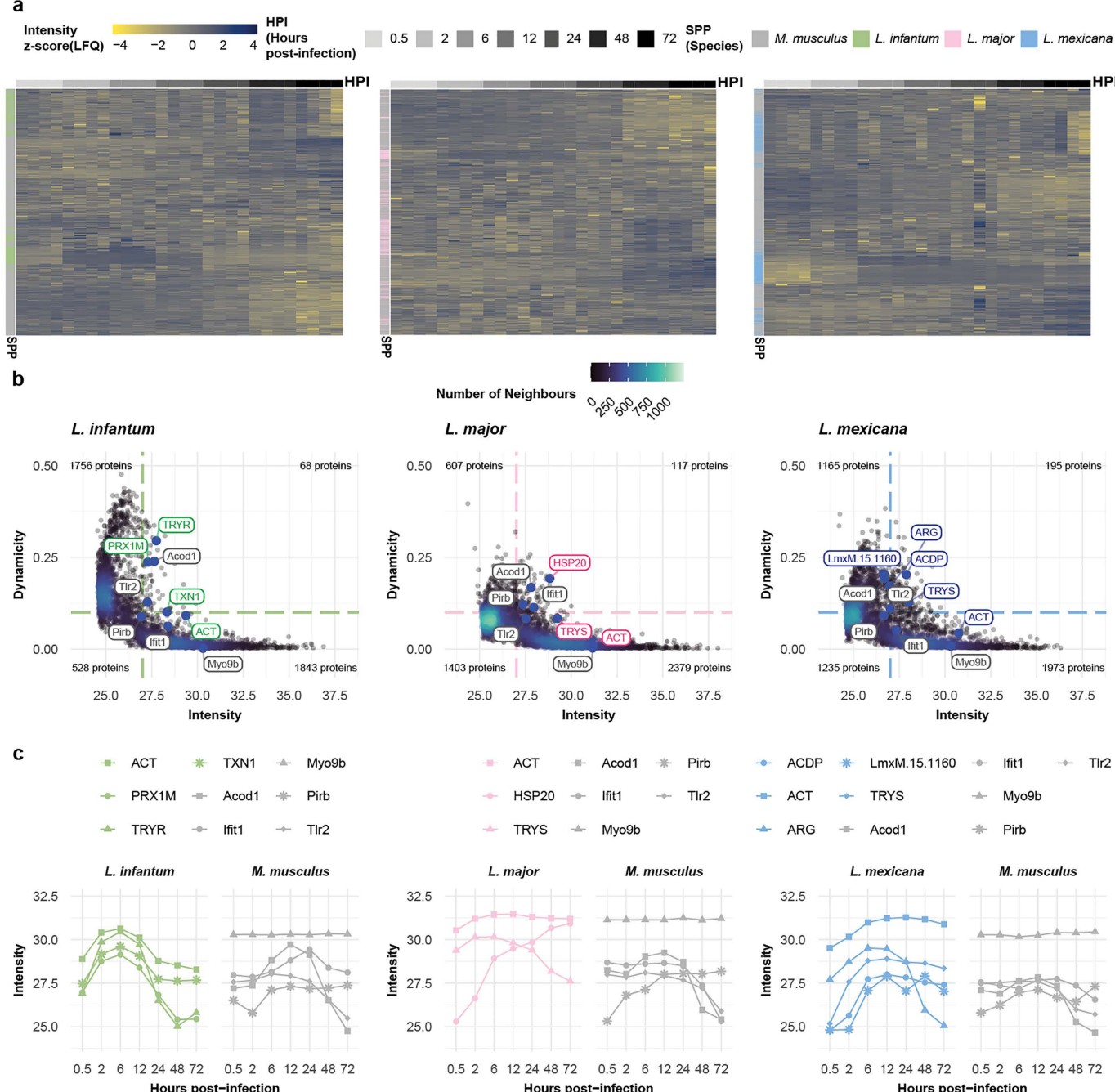

**Fig 2. Proteome dynamics reveal different protein expression profiles for the three *Leishmania* spp. experiments. a,** Heatmap of quantified proteins for the *L. infantum* (green), *L. major* (pink), and *L. mexicana* (blue) experiments. Each row represents the abundance (z score [LFQ]; yellow-to-blue scale) of each quantified protein across the postinfection timepoints (grayscale). Proteins are annotated as either *M. musculus* (gray) or belonging to *Leishmania* spp. (green, pink, and blue for *L. infantum*, *L. major*, and *L. mexicana*, respectively). **b,** Scatter plot of quantified proteins for the *L. infantum* (green), *L. major* (pink), and *L. mexicana* (blue) experiments. Each dot represents the abundance (mean [log2]; x-axis) and dynamicity (Gini score; y-axis) of each quantified protein. The colors of the dots reflect the density distribution (bluescale). Highlighted dots (larger, blue) represent known dynamic and stable *M. musculus* (gray) or *Leishmania* spp. (green, pink, and blue for *L. infantum*, *L. major,* and *L. mexicana*, respectively) proteins. **c,** Line plot of highlighted proteins for the *L. infantum* (green), *L. major* (pink), and *L. mexicana* (blue) experiments. Each dotted line represents the protein abundance (mean [log2]; y-axis) across the hours postinfection (x-axis). Dotted lines are color coded using grayscale for *M. musculus* proteins and greenscale, pinkscale, or bluescale for *L. infantum*, *L. major*, and *L. mexicana* proteins, respectively.

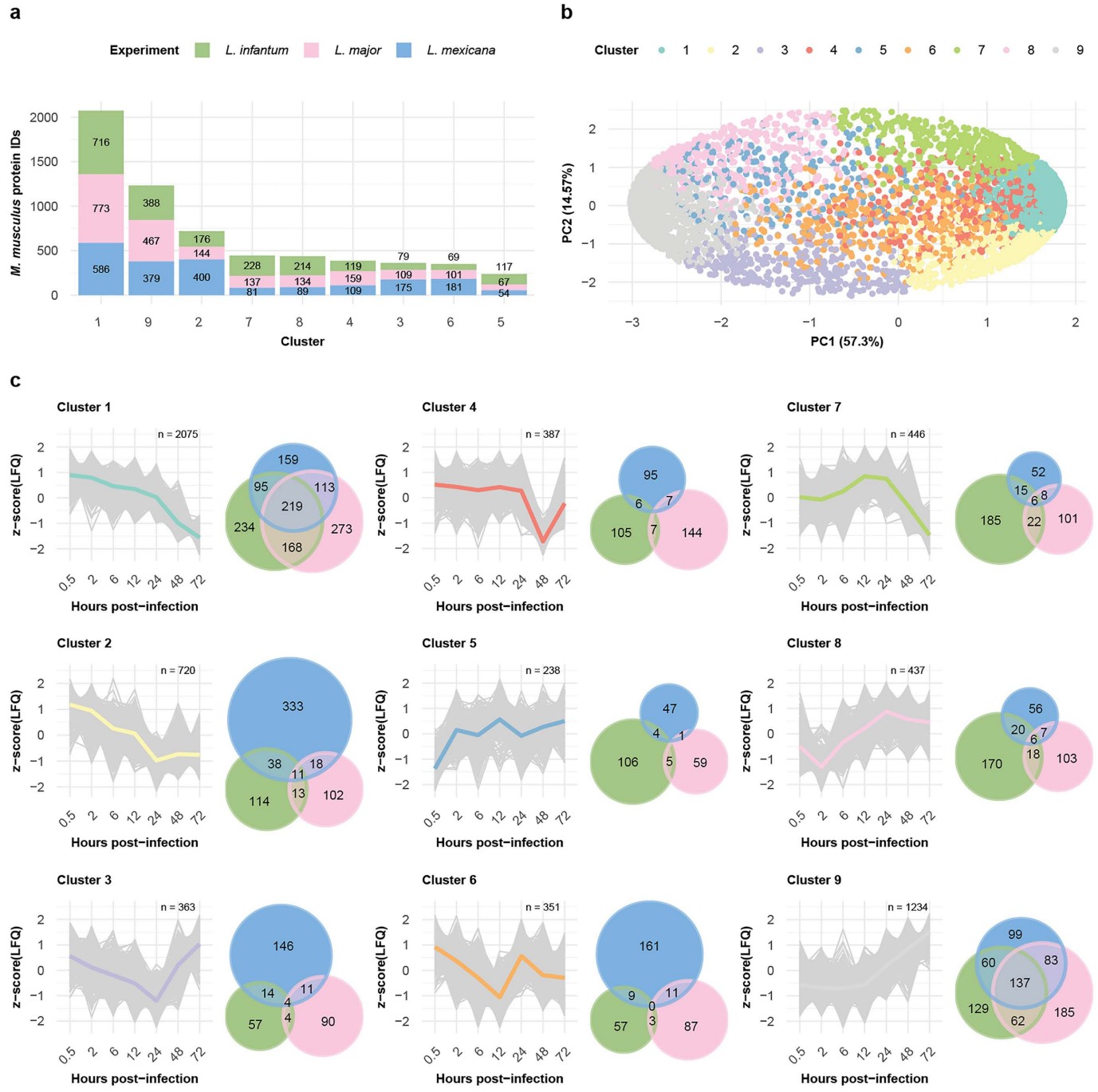

**Fig 3. Clustering analysis revealed differences and similarities between the expression profiles of the *M. musculus* proteins. a,** Bar plot displaying the number of *M. musculus* proteins within each self-organizing map (SOM) cluster. The fractions of protein per bar from the *L. infantum* (green), *L. major* (pink), and *L. mexicana* (blue) experiments are indicated. **b,** Scatter plot showing the first two components of the principal component analysis (PCA), which together explain 81.87% of the data variance. The variance explained by each principal component is labeled on its respective axis. Protein IDs are depicted as dots and colored on the basis of their SOM cluster assignment. **c,** SOM cluster panel. The line plot depicts the protein abundance (z score [LFQ]; y-axis) across the hours post infection (x-axis) for each protein ID (gray) assigned to the cluster. The mean abundance of a cluster is represented by its designated color. An adjacent Venn diagram depicts the overlap between the *M. musculus* protein IDs from the *L. infantum* (green), *L. major* (pink), and *L. mexicana* (blue) experiments assigned to the cluster.

by *L. mexicana* (Fig 4A and 4C and S7 Table). The *L. infantum*-dominated clusters 2 and 5 presented very distinctive expression profiles, with expression peaking at either 6 hpi or 48 hpi. Cluster 4, whose expression peaked at 48 hpi, was dominated by *L. major* proteins. Cluster 8, with a majority of *L. mexicana* protein groups, presented consistently high expression at 6 hpi. Three clusters (clusters 3, 6 and 9) presented similar expression patterns (Fig 4C): protein abundance increased, followed by a plateau, and finally decreased with a time difference in the plateau phase. For Cluster 3 (*L. major* dominated), the plateau spans from 2 hpi to 12 hpi; for Custer 6 (mostly shared between *L. infantum* and *L. major*), it spans from 2 hpi to 6 hpi; and for Cluster 9 (*L. mexicana* dominated), it spans from 6 hpi to 12 hpi. To further investigate the functional annotations of the proteins within the clusters, we queried the GO database and found several overrepresented terms among the clusters (S9 Fig and S8 Table).

Overall, we observed different temporal protein expression patterns for the *Leishmania* spp., with many protein profiles peaking earlier in *L. infantum* than in *L. major* and *L. mexicana*.

## Shortlisting of virulence factor candidates

To further explore our dataset, we determined the number of significantly up- or downregulated proteins between adjacent time points (P value ≤0.05 and absolute fold change ≥2, $c = 0.05$; Fig 5A and S9, S10 and S11 Tables). We observed that the greatest number of significantly upregulated proteins for *L. infantum* and *L. major* occurred between 0.5 hpi and 2 hpi (230 and 58 proteins, respectively), whereas for *L. mexicana*, this widespread upregulation occurred between 6 hpi and 12 hpi (246 proteins). Conversely, the greatest number of significantly downregulated proteins for all three *Leishmania* spp. was detected between 12 hpi and 24 hpi (164 proteins for *L. infantum*, 34 for *L. major*, and 26 for *L. mexicana*). This mirrored the plateau clusters observed during the SOM analysis, where *L. mexicana* also exhibited a temporal delay in major proteome remodeling compared with *L. infantum* and *L. major* (Fig 4C). For the host response, the greatest number of up- and downregulated proteins was found at the same time point between 24 hpi and 48 hpi (Fig 5A and S7 and S8 and S9 Tables).

We overlapped all significantly up- and downregulated proteins (Fig 5B) to identify proteins commonly regulated in all three infection time courses. Among the 23 *Leishmania* spp. proteins were TRYR and AGC essential kinase (OG6_105018), while the *M. musculus* proteins included the key inflammatory markers Acod1 and Tlr2 and the tubulin protein Tuba1c. Despite being significantly regulated in all three species, these proteins still presented different expression profiles (Fig 5C), with a delay in peak expression for *L. mexicana* as the most distinctive feature. From the pool of 23 shared *Leishmania* spp. proteins, we selected a protein annotated as a glycosomal membrane protein (LmxM.28.2260), whose orthologs were previously investigated in *Trypanosoma brucei* and *T. cruzi* [40–42]. Additionally, we selected LmxM.10.0130, a hypothetical protein that is significantly regulated only in *L. mexicana*.

## Deletion of putative *L. mexicana* virulence factors resulted in reduced infectivity of BMDMs

To investigate the functional significance of our selected candidate proteins for efficient infection of *L. mexicana*, we employed CRISPR-Cas9 editing to knockout two genes (LmxM.10.0130 and LmxM.28.2260). Infection rates were determined by counting macrophages containing mNG-labeled parasites. We also assessed the differentiation ability of knockout cell lines under axenic conditions. Additionally, we tagged the genes endogenously with mCherry to analyze their cellular localization.

Available information in online databases on LmxM.28.2260 describes it as a glycosomal membrane protein with a size of 24 kDa and lists over 70 genes in its ortholog group (OG6_142694) [43]. In *T. brucei* and *T. cruzi,* the protein is described as peroxisomal biogenesis factor 11 (PEX11), and the TrypTag project localized it in glycosomal compartments of *T. brucei* procyclic cells[43]. We tagged the gene in a cell line expressing GFP-GAPDH, an established marker for glycosomal compartments in trypanosomatids, to investigate the localization of our candidate. Our microscopy images

PLOS Pathogens

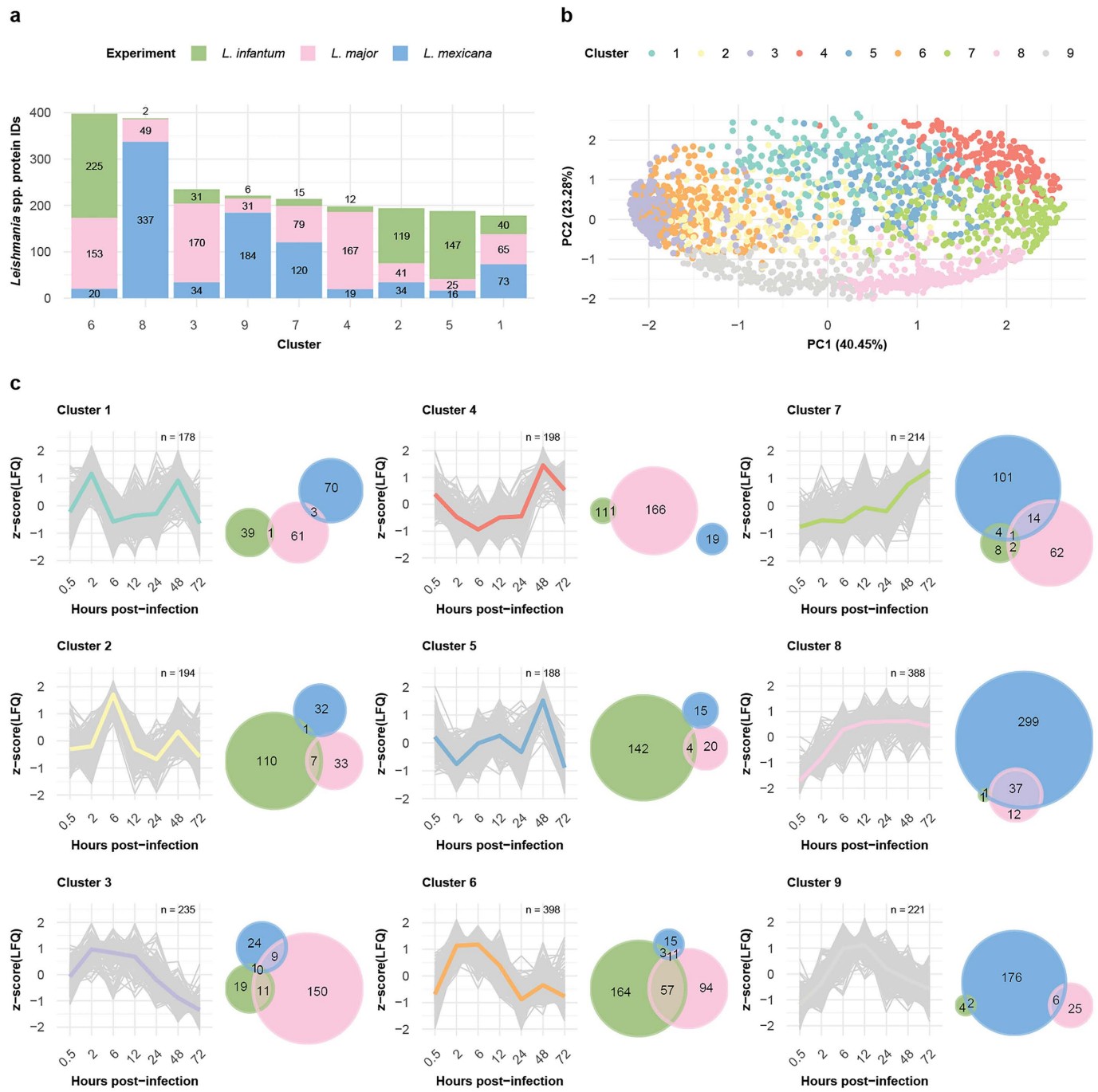

**Fig 4. Clustering analysis revealed differences and similarities between the expression profiles of the *Leishmania* spp. proteins. a,** Bar plot displaying the number of *Leishmania* spp. proteins within each self-organizing map (SOM) cluster. The fractions of *L. infantum* (green), *L. major* (pink), and *L. mexicana* (blue) proteins per bar are indicated. **b,** Scatter plot showing the first two components of the principal component analysis (PCA), which together explain 63.73% of the data variance. The variance explained by each principal component is labeled on its respective axis. Protein IDs are depicted as dots and colored on the basis of their SOM cluster assignment. **c,** SOM cluster panel. The line plot depicts the protein abundance (z score [LFQ]; y-axis) across the hours post infection (x-axis) for each protein ID (gray) assigned to the cluster. The mean abundance of a cluster is represented by its designated color. An adjacent Venn diagram depicts the overlap between the OrthoMCL IDs assigned to *L. infantum* (green), *L. major* (pink), and *L. mexicana* (blue) *Leishmania* spp. protein IDs in the cluster.

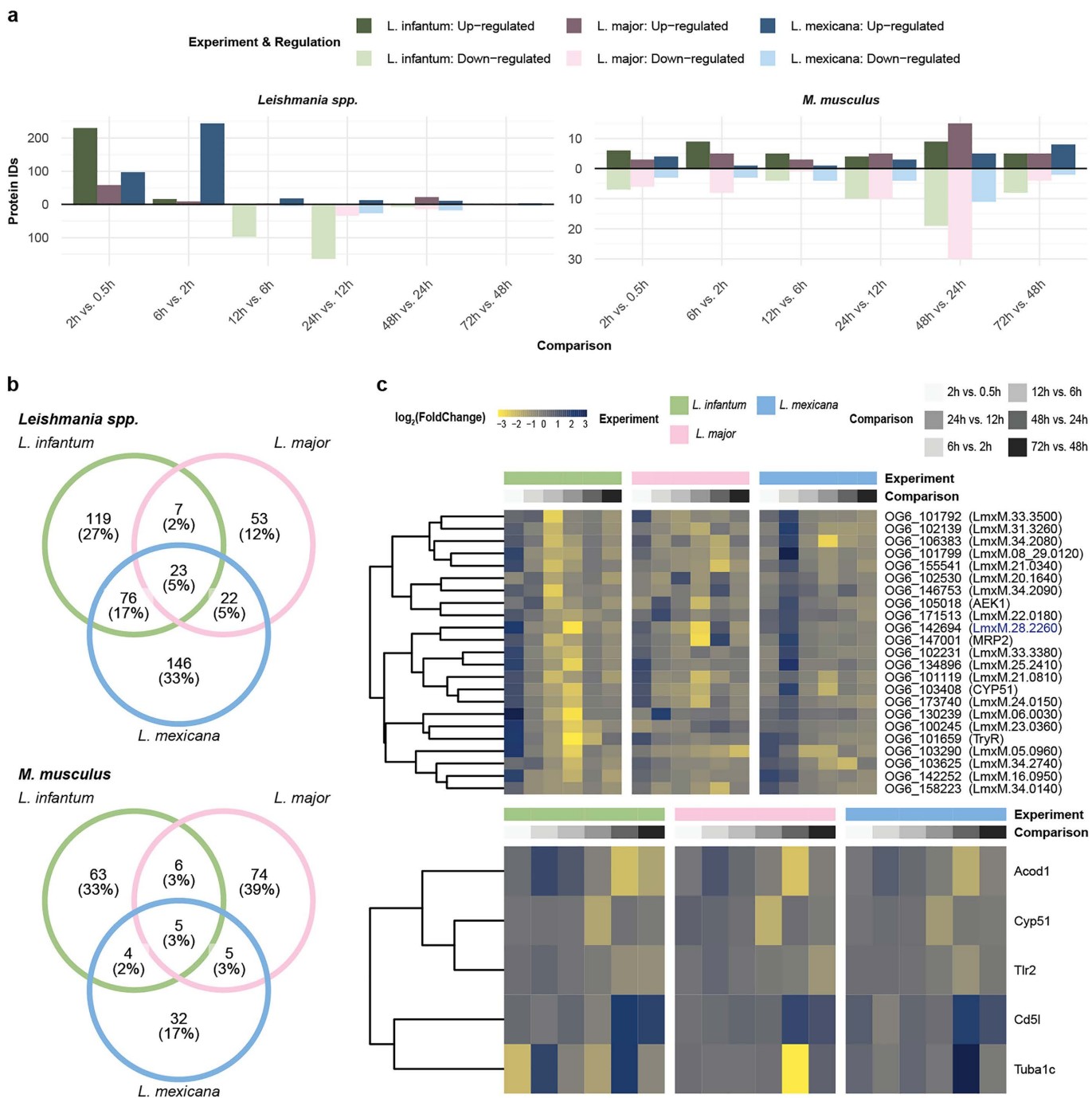

**Fig 5. Pairwise differential expression analysis revealed *Leishmania* spp. and *M. musculus* protein IDs with significantly modulated expression profiles. a,** Bar plot showing the number of significantly up- or downregulated *Leishmania* spp. (right panel) and *M. musculus* (left panel) proteins (p value < 0.05 and abs[log2(FoldChange)] > 1) for the *L. infantum* (green), *L. major* (pink), and *L. mexicana* (blue) experiments. **b,** Venn diagram depicting the overlap between OrthoMCL IDs assigned to the significant *Leishmania* spp. protein IDs (upper panel) and the significant *M. musculus* protein IDs (lower panel) from the *L. infantum* (green), *L. major* (pink), and *L. mexicana* (blue) experiments. **c,** Heatmap displaying the OrthoMCL IDs assigned to the significant *Leishmania* spp. protein IDs (upper panel) and the significant *M. musculus* protein IDs (lower panel) from the *L. infantum* (green), *L. major* (pink), and *L. mexicana* (blue) experiments. Each row represents the abundance (z score [LFQ]; yellow-to-blue scale) of each quantified protein across pairwise hours after infection (grayscale).

revealed that the LmxM.28.2260 signal clearly overlapped with that of GAPDH (Fig 6A) in both promastigotes and axenic amastigotes.

Although the expression patterns of LmxM.28.2260 and its orthologs in *L. major* and *L. infantum* were significantly modified over all three time courses, we observed a clear difference in their expression patterns (Fig 6B). In *L. mexicana,* the protein was upregulated continuously until 24 hpi and then reached a plateau during the further course of infection. In *L. infantum* and *L. major,* we measured an increase in protein levels during early time points of infection, with a peak at 2–6 hpi. Afterwards, the protein levels decreased in both species. This finding was consistent with our previous observation of species-specific protein expression dynamics during the course of infection (Fig 4) and also correlates with the dynamic of infection profiles of the three parasite species (S1 Fig).

We also evaluated the role of our candidate in stage differentiation from promastigotes to amastigotes. To that end, we differentiated stationary phase promastigotes under axenic conditions into amastigotes. Compared with the WT cell line, the knockout of LmxM.28.2260 resulted in a reduced cell density after 48 h (Fig 6C). Complementation of the protein through ectopic expression in the knockout cell line restored differentiation ability, implicating a function of our candidate protein in the differentiation process from promastigotes to amastigotes. We also tested the growth of promastigotes and measured a slightly reduced population doubling time of 7–8 h in the knockout cells compared with the WT cells (6–7 h) (S5 Fig).

Finally, we investigated the role of our candidate in the infection of BMDMs. Infection experiments with the knockout cell lines revealed a marked decrease in infection rates compared with those of the WT. At 24 hpi, the infection rates of the ΔLmxM.28.2260 cell line dropped below 40% and further declined to 20% after 72 hours. In contrast, the WT maintained infection rates between 50% and 60% (Fig 6D). The rescue cell line also presented significantly higher infection rates than did the knockout cell line at 48 hpi. However, we could not fully restore the infection rates of the WT parasites.

To investigate the role of LmxM.10.0130 in the infection process, we conducted the same set of experiments described above. Available information in databases was more limited than that in the first candidate. TriTrypDB described LmxM.10.0130 as a hypothetical protein with a predicted size of 79.3 kDa, with over 70 proteins belonging to the same ortholog group (OG6_146547) [43]. The TrypTag project was able to tag orthologs in procyclic *T. brucei* cells, revealing their localization to the posterior tip, flagellar pocket, and endocytic and glycosomal (<10%) compartments [44]. We compared the fluorescent signals of our candidate in our glycosomal marker cell line. The fluorescence signal of LmxM.10.0130 was irregularly distributed in the cell body, with relatively high intensities adjacent to the kinetoplast in promastigotes (Fig 6E). We did not observe a distinctive overlap with signals from glycosomal compartments in neither promastigotes nor amastigotes. The proximity of the LmxM.10.0130 signal to the kinetoplast could indicate an association with early endosomal compartments.

Although *L. infantum* and *L. major* orthologs were also detected and quantified, we only measured a significant difference between time points in *L. mexicana.* Here, we noted that protein levels remained low until 72 hpi (Fig 6F). As a differentiation marker, we measured the protein levels of paraflagellar rod protein 2 (PFR-2), an important structural protein in the flagellum [45]. In amastigotes, the flagellum is shortened drastically and barely extends out of the cell body; hence, a decrease in PFR-2 protein levels is expected. Since we observed an inverse expression profile of PFR-2 and LmxM.10.0130, we also investigated the differentiation ability of ΔLmxM.10.0130 cells. We observed a clear difference compared with the WT cells (Fig 6G). The knockout cells were unable to proliferate as axenic amastigotes, although they had the morphological characteristics of amastigotes, such as a rounded cell body and a shortened flagellum (S10 Fig). Here, reintroduction of the LmxM.10.0130 gene fully restored differentiation capacity. In promastigotes, knockout of LmxM.10.0130 resulted in a slightly prolonged population doubling time of 7–8 h.

In infection experiments in BMDMs, compared with WT cells, ΔLmxM.10.0130 cells presented a lower infection rate after 72 h, which is in line with the regulatory profile (Fig 6H). Complementation significantly increased the infectivity of the cells at 72 hpi, but infection rates were not fully restored to WT levels.

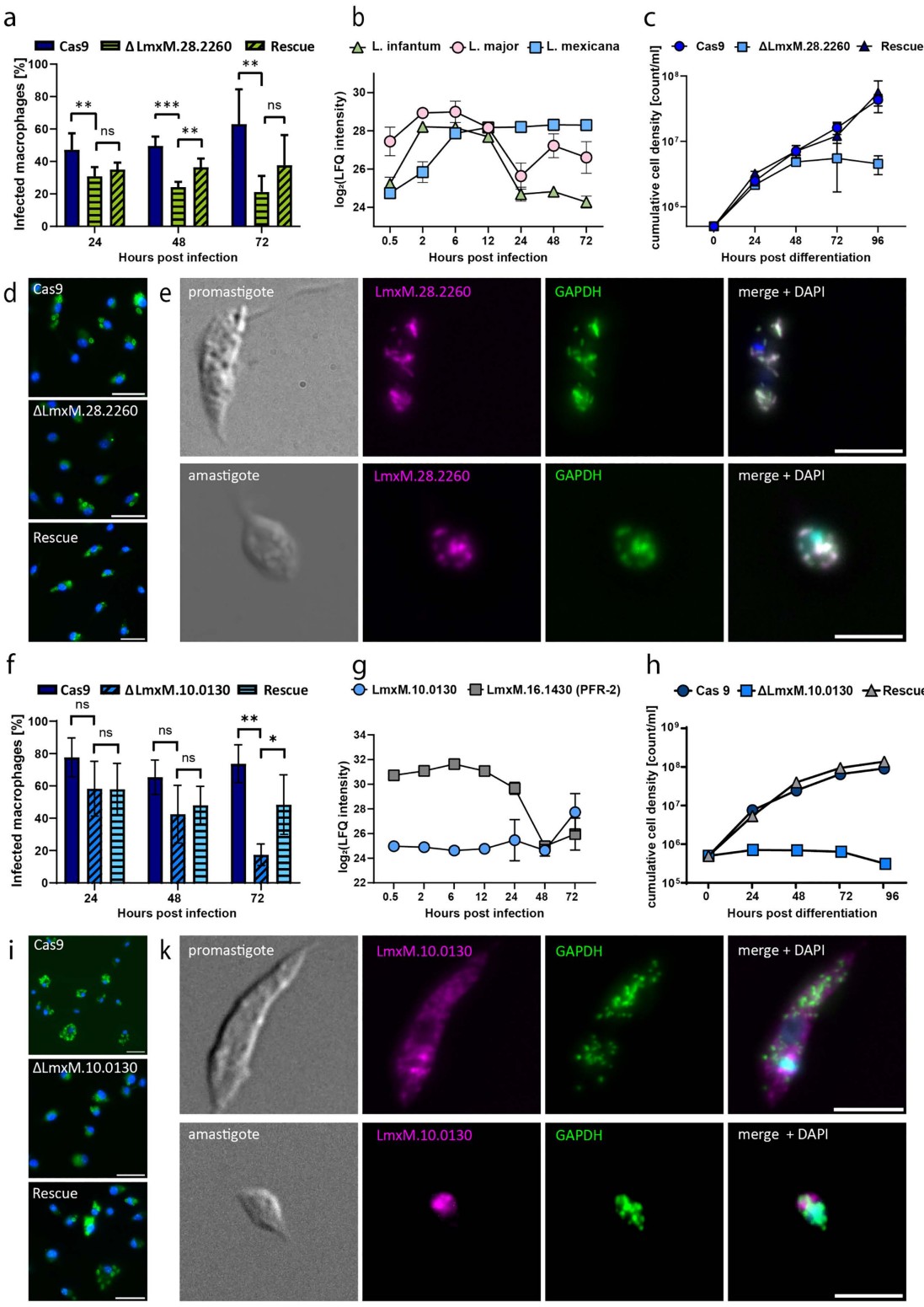

**Fig 6. Knockout of two upregulated *L. mexicana* proteins results in reduced infectivity and a distinct differentiation phenotype. a,** Fluorescent signal of endogenously tagged LmxM.28.2260 overlapping with the glycosomal marker (GAPDH) signal in both promastigotes and amastigotes (scale bar: 5 µm). **b,** Label-free quantification values for LmxM.28.2260 and orthologs in *L. major* and *L. infantum* show species-dependent relative protein

levels throughout the infection. **c,** Axenic differentiation to amastigotes is affected by knockout of LmxM.28.2260 (blue square), with a cell density not exceeding 6*10⁶ c/ml. Rescue of the knocked out gene (blue triangle) re-established the growth phenotype to WT conditions (blue circle). **d,** Infection experiments with ΔLmxM.28.2260 in BMDMs revealed significantly reduced infectivity compared with that of the positive control (Cas9) starting at 24 h (unpaired t test, p < 0.05). The rescue cell line partially restored the infection phenotype of the positive control after 48 h (unpaired t test, p < 0.05). A significant difference in the infection rate between the rescue and knockout cell lines was not detected after 72 h. **e,** The fluorescent signal of endogenously tagged LmxM.10.0130 (magenta) does not overlap with the GAPDH signal (green) in promastigotes or amastigotes (scale bar: 5 µm). The LmxM.10.0130 signal is irregularly dispersed throughout the cell body, with a relatively high intensity near the kinetoplast. **f,** Relative protein levels of LmxM.10.0130 were low throughout infection and increased only at 72 h. In comparison, the protein levels of PFR-2 were high until 24 h and then decreased. **g,** LmxM.10.0130 knockout cells are unable to differentiate under axenic conditions (blue square) when the cell density does not exceed 10⁶ c/ml. Rescuing the knocked out gene (gray triangle) enables the cells to reestablish their differentiation phenotype to WT conditions (blue circle). **h,** Infection experiments with ΔLmxM.10.0130 revealed a significant difference in infection rates compared with the positive control (Cas9) at 72 h (unpaired t test, p < 0.005). The rescue cell line partially restored the infection phenotype of the positive control, which was significantly different from that of the knockout cell line.

In summary, we provide a comprehensive quantitative mass spectrometry analysis of the infection process of macrophages with three different *Leishmania* species. As a proof of principle for functional significance during infection, we showed that two *L. mexicana* proteins from our proteomic dataset are critical for establishing infection in BMDMs. The differentiation of promastigotes to amastigotes was affected by the absence of proteins, which might contribute to the infection phenotype.

## Discussion

Through proteome profiling, we were able to identify putative novel virulence factors involved in *Leishmania* infection of BMDMs. Our large experimental approach allowed us to uncover both similarities and key differences during infection with three *Leishmania* spp., which may help in understanding disease pathology and identifying novel proteins involved in the infection process of leishmaniasis. Our *Leishmania* infectome database (LInfDB) is available through an online interactive platform (https://butterlab.imb-mainz.de/LInfDB).

The identification of many *M. musculus* proteins that are shared across the time courses, suggests that a core set of macrophage proteins is involved in the host response to *Leishmania* infections, independent of the species with which they are infected (Figs 1B and 3). However, for the *Leishmania* species, differences in protein expression dynamics during infection were observed (Figs 2C and 4). These findings support the notion that disease pathology is driven mainly by the parasite and not the host. Previous studies using RNA sequencing to identify human host and pathogen transcripts have yielded comparable results [46]. We confirmed that while the macrophages elicited a parasite response, they did not exhibit a species-specific response. However, although macrophages might be long-term host cells for parasites, other mononuclear phagocytes, such as dendritic cells and neutrophils, might also play important roles in the persistence and spread of parasites in the host [7]. There is active discussion among parasitologists and immunologists on the importance of the host's response in disease pathology. With our dataset, we not only show the direct dynamic response of both the parasite and the host during infection but also highlight the diverse responses of the parasite in the infection process.

Our dataset provides valuable information for potential new drug targets, as more than half of the significantly regulated *Leishmania* proteins (Fig 5) are annotated to be enzymatically active. Among these proteins is TRYR, which has already been intensively investigated as a potential drug target to treat leishmaniasis and other trypanosomatid diseases [47–48]. In our proof-of-principle experiments, we demonstrated that two *L. mexicana* proteins are essential for the infection process (Fig 6A and 6E). In both cases, CRISPR-Cas9-mediated depletion of the proteins resulted in reduced infection of BMDMs. In particular, the experiments with LmxM.10.0130, a hypothetical protein, highlight the usability of our dataset. Here, we describe a protein previously unknown to be involved in the infection process of *L. mexicana*. Experiments with this protein also underscore the differences in the infection dynamics among the three species, as this protein was significantly regulated only in *L. mexicana* but not in *L. infantum* or *L. major*. Our experimental data with LmxM.28.2260 provide

additional insight into the differences in the infection dynamics of the three *Leishmania* species. LmxM.28.2260 was significantly regulated in all three species, but the dynamics of the protein differed from species to species (Fig 6B). The ortholog of LmxM.28.2260 in *T. brucei* (*Tb*PEX11) has been implicated in the proliferation of peroxisomes and has been proven to be essential for the survival of *T. brucei* [40–49]. In contrast, in *L. mexicana,* the protein is not essential in pro-mastigotes but is essential for developmental differentiation to amastigotes. In *T. cruzi,* the ortholog (TcCLB.509203.40) is localized to its contractile vacuole (CV) [42]. CVs play essential roles in cell volume regulation [50–52], Ca2 + homeostasis [53] and the transport of proteins to the plasma membrane surface of *T. cruzi* [54], but the exact role of TcCLB.509203.40 in this cellular compartment is still not known. Given the differences in the protein dynamics of LmxM.28.2260 and its orthologs in *L. major* and *L. infantum,* it is possible that the protein may play species-specific roles in the infection process. However, further loss-of-function studies are needed to clarify potential distinct roles of these proteins across different *Leishmania* species.

As climate change also affects the spread of the *Leishmania* vector, leading to an expansion of endemic regions such as the Mediterranean basin, rapid diagnostic tools have become more important. Clinical presentation alone is not sufficient to monitor the spread of leishmaniasis, as studies have shown that asymptomatic infections occur frequently [55–59]. Here, our dataset also offers new targets for diagnostics, as protein dynamicity (Fig 2B and S3 Table) is unique to each of the three species. Most of the parasite proteins with high dynamicity and protein expression did not exhibit the same dynamicity and intensity scores in the other two *Leishmania* species.

In summary, we believe that our dataset provides a valuable resource in the worldwide effort to fight leishmaniasis. We offer new insights into the infection dynamics of *Leishmania*, with two novel virulence factors for *L. mexicana*, and the potential to identify and investigate new drugs and diagnostic targets.

## Methods

### Cultivation of *Leishmania* parasites

*Leishmania mexicana* (MNYC/BZ/62/M379), *L. major* (MHOM/IL/81/FEBNI) and *L. infantum* (MHOM/00/98/LUB1) pro-mastigotes were cultivated in Schneiders Drosophila Medium (SDM, Sigma, S9895) supplemented with 2% urine, 10% FCS, 4% AAP mix (50 ml, Pen/Strep, 50 ml 100x sodium pyruvate, 50 ml 100x L-glutamine, 50 ml DMEM w/o phenol red, 0.18 g L-asparagine, 0.58 g L-arginine) and 1% HEPES (buffered with KOH, pH 6.9). The logarithmic growth of the cells was maintained (up to $1*10^7$ c/ml) at 28°C and 5% $CO_2$. Antibiotics were added to the cell lines at the indicated concentrations. *For in vitro* differentiation of *L. mexicana* promastigotes to axenic amastigotes, logarithmically growing promastigotes were diluted to $1x10^5$ cells/ml in a total volume of 3 ml SDM and further cultivated at 28°C and 5% $CO_2$ for five days. To let cells grow into the stationary phase during this time period, they were counted daily, but not diluted. After five days, cells were diluted to $3x10^6$ c/ml in 20 ml Differentiation medium (SDM (Gibco) further supplemented with 25 µg/ml gentamicin, 20% heat-inactivated FCS, 25 mM 2-(N-morpholino)-ethanesulfonic acid (MES), pH5.4) and cultivated for two days at 32°C and 5% $CO_2$.

### Generation of bone marrow-derived macrophages

The bone marrow of WT C57BL/6 mice was harvested as described previously [60]. In brief, bone marrow was isolated from the femur and tibia by flushing via a 26G syringe filled with bone marrow cultivation medium (DMEM [Gibco, 31966], Pen/Strep, 10% FCS, 5% horse serum, HEPES, and 5% nonessential amino acids [Gibco, 11140050]). The bone marrow suspension was then centrifuged at 250xg for 10 min, and the pellet was resuspended in 5 ml of cultivation medium. Differentiation of bone marrow cells was carried out in a 50 ml volume by adding 1 ml of cell suspension to 44 ml of cultivation medium. Differentiation was induced by adding 5 ml of macrophage colony stimulating factor (M-CSF) and incubating it for 7 days at 37°C and 5% CO2. M-CSF was harvested from the L929 culture supernatant as described previously [61].

## Infection of BMDMs

BMDMs were infected with stationary promastigotes (108 c/ml) at an MOI of 5 and incubated for 4 h. Afterwards, the BMDMs were washed 3 times with prewarmed PBS and then incubated at 37°C and 5% $CO_2$ in cultivation medium until each respective timepoint was harvested.

## Mass spectrometry sample preparation

Protein extracts of mouse macrophages infected with different *Leishmania* spp. (10 µg in 30 µl of LDS + 10 mM DTT) were heated for 10 minutes at 70°C. Proteins were then separated on a 10% NOVEX gradient SDS gel (Thermo Scientific) for 8 min at 180 V in 1x MES buffer (Thermo Scientific). The proteins were fixed, stained with a Coomassie blue solution (0.25% Coomassie blue G-250 [Carl Roth], 10% acetic acid, 43% ethanol) and washed with water. Each gel lane was cut into three slices, minced, and destained with a 50% ethanol/50 mM ammonium bicarbonate pH 8.0 solution. Proteins were reduced in 10 mM DTT for 1 h at 56°C and then alkylated with 50 mM iodoacetamide for 45 min at room temperature in the dark. Proteins were digested with MS-grade trypsin (Sigma) overnight at 37°C. Peptides were extracted from the gel twice via a mixture of 30% acetonitrile and 50 mM ammonium bicarbonate, pH 8.0, and three times via pure acetonitrile, which was subsequently evaporated in a concentrator (Eppendorf) and loaded onto activated C18 material (Empore) StageTips as previously described [62].

## Mass spectrometry data acquisition and analysis

Peptides were separated on a 50 cm long, 75 µm inner diameter column (New Objective) self-packed with ReproSil-Pur 120 C18-AQ (Dr. Maisch GmbH) by reverse-phase chromatography. The EASYnLC 1200 (Thermo Fisher) was mounted to an Orbitrap Exploris 480 plus mass spectrometer (Thermo Fisher), and peptides were eluted from the column in an optimized 88-minute gradient from 2 to 40% of 80% MS-grade acetonitrile/0.1% formic acid solution at a flow rate of 250 nl/min. The mass spectrometer was operated in positive ion mode with a data-dependent acquisition strategy of one MS full scan (scan range 300–1,650 m/z; 60,000 resolution; normalized AGC target 300%; max IT 28 ms) and up to twenty MS/MS scans (15,000 resolution; AGC target 100%, max IT 40 ms; isolation window 1.4 m/z) with peptide matching preferred via high-energy collisional dissociation (HCD) fragmentation. The raw MS data were searched via the Andromeda search engine [63] integrated into MaxQuant suite 1.6.5.0 [64] via Ensembl [65] *M. musculus* reference proteome sequences (68,381 entries), TriTrypDB [37] *L. infantum* (TriTrypDB-50_LinfantumJPCM5_AnnotatedProteins.fasta; 8,591 entries), *L. major* (TriTrypDB-50_LmajorFriedlin_AnnotatedProteins.fasta, 8,519 entries) and *L. mexicana* (TriTrypDB-50_LmexicanaMHOMGT2001U1103_AnnotatedProteins.fasta, 8,246 entries) reference proteomes for each respective experiment. Carbamidomethylation at cysteine was set as a fixed modification, and methionine oxidation and protein N-acetylation were set as variable modifications. The match between runs option was activated.

## Bioinformatics analysis

For protein quantification, contaminants, reverse database hits, protein groups identified only by site, and protein groups with fewer than 2 peptides or with nonunique peptides were removed from the MaxQuant proteingroups.txt file. Missing values were imputed by shifting a beta distribution, derived from the label-free quantification (LFQ) intensity values, to the limit of quantitation. Further analysis and graphical representation were conducted within the R framework (R Core Team 2021) incorporating, among other packages, ggplot2 [66] for visualization. For exploratory analysis of the proteomic data, we used base R functions. Principal component analysis (PCA) was performed using the prcomp function to visualize patterns of variance across samples. Additionally, Euclidean distances between samples were calculated using the dist function to assess overall similarity and clustering in the dataset. For protein dynamicity analysis, protein abundance stability across post-infection timepoints was evaluated via the Gini coefficient. The coefficient (G) is calculated for each

protein as the mean of the difference between every possible pair of timepoints, divided by the mean size (μ) with the following formula:

$$G = \frac{\sum_{i=1}^{n} \sum_{j=1}^{n} \lfloor x_i - x_j \rfloor}{2n^2 \mu}$$

For the self-organizing map (SOM) analysis, host and parasite datasets were analysed independently. The number of clusters (nine per species) was determined empirically, based on both intracluster distances (i.e., similarity of protein expression profiles within clusters) and neighborhood distances between clusters to ensure meaningful separation. Proteins below the 25th percentile of LFQ intensity, and subsequently, those below the 10th percentile of the LFQ intensity interquartile range, were excluded prior to clustering. The SOM clustering algorithm was applied as implemented in the Kohonen R package [67].

For pairwise time point comparisons, the protein enrichment threshold was set to a p value ≤ 0.05 (Welch's t test) and an absolute fold change ≥ 2, c = 0.05. Welch's t test was performed with 4 replicates. For functional enrichment analysis, proteins were queried in the Gene Ontology database [68] via the ClusterProfiler R package [69]. Terms found among the enriched proteins were tested for overrepresentation with an adjusted P value (false discovery rate) ≤0.05 (Fisher's exact test) against all terms found in the background (defined as all quantified proteins in the comparison, whether enriched or not).

## Transfection of *L. mexicana*

Tagging and knockout DNA sequences were amplified from pPLOT plasmids [70] and primers for genes were designed using the corresponding LeishGEdit website. Templates for the sgRNAs were amplified with the G00 primer and corresponding primers for the 5' or 3' insertion of the constructs into the target site. Amplified constructs were checked for the correct size via agarose gel electrophoresis (0.8% agarose gel, 120 V, 25 min), and tagging/knockout constructs were purified with isopropanol. *L. mexicana* cells were transfected via electroporation using transfection buffer described in [71] and the Amaxa Nucleofector IIb with the "X-001 free choice" transfection program. *Leishmania* cells were transfected during the logarithmic growth phase. A total of $2*10^7$ cells were transfected in a total volume of 400 μl with 5 μg of donor DNA (PCR product) and 20 μl of sgRNA template. Knockout cell lines were transfected with 10 μg of rescue construct (plasmid). After electroporation, the cells were transferred to 10 ml of culture medium and incubated for 16 h. After 16 h, antibiotics were added, and the cells were diluted 1:10 and 1:100 in a 96-well plate. The cells were then incubated for 10–14 days until single clones could be picked from individual wells.

## Cloning of the rescue construct

Rescue constructs were subsequently cloned and inserted into pRM005 plasmids[69]. In brief, the plasmid was digested for 2 h at 37°C using EcoRI and SpeI restriction enzymes. The open reading frame (ORF) of the gene of interest was amplified from genomic *L. mexicana* DNA. Genomic DNA was isolated via the Roche High Pure PCR Template Preparation Kit (Roche, 11796828001). The forward and reverse primers were designed so that the annealing site was attached to the 5' or 3' end of the ORF. A double Ty-Tag was added through the forward primer.

## Immunofluorescence analysis of macrophages and parasites

The macrophages on coverslips were fixed in 200 μl of PBS containing 4% paraformaldehyde (PFA) (10 min, RT). The cells were washed three times with 500 μl of PBS. The cells were permeabilized in 0.5% PBS-Triton (20 min, RT, dark) and then washed twice in 0.1% PBST (1 h, RT, dark). The cells were then mounted in Fluoromount + DAPI. Images were captured with a Leica DMI 6000B microscope and processed with Fiji software. *Leishmania* cells were fixed by harvesting

2*106 cells (1500xg, 5 min, RT) and washing once in PBS. The parasites were then fixed in 200 µl of 4% PFA (10 min, RT). The cells were then washed three times in 500 µl of PBS (1500xg, 5 min, RT) and added to silianized coverslips via centrifugation (1000xg, 2 min, RT). For immunofluorescence, the parasites were permeabilized in 0.5% PBST and then blocked with 1% BSA in PBS. Primary antibodies were added at the appropriate dilutions in 0.2% PBST (1 h, RT). The cells were then washed 3x with 0.1% PBST (RT, 3 min). The secondary antibodies were added at their respective dilutions in 0.1% PBST (45 min, RT, dark). The cells were again washed 3x in 0.1% PBST (RT, 3 min). Coverslips were mounted on microscopy slides with Fluoromount + DAPI and imaged using a Leica Thunder Imager (DMi8, Leica Microsystems, Wetzlar).

## Evaluation of infected macrophages via the ImageJ macro

The percentage of infected macrophages was determined via fluorescence microscopy. Fluorescent *L. mexicana* cells (mNG::ß-tub) were used to infect macrophages. The samples were prepared and imaged as described above. A Leica application suite tile scan was used to image a 0.35–0.42 mm$^2$ region, and 3 regions were imaged per coverslip. The total number of macrophages was determined by their DAPI signal, and infected macrophages were counted by determining the number of mNG signals directly adjacent to the DAPI signals. If the mNG fluorescent signal was not strong enough or if the background was too high, infected macrophages were counted by hand. Infection experiments were carried out in triplicate with BMDMs obtained from three different mice.

## Scanning electron microscopy

Briefly, 1x10$^7$ *L. mexicana* cells per sample were harvested (1,000 g, 3 min, RT), and the supernatants were removed, except for a few microliters. The cells were fixed by the addition of 900 µl of prewarmed (27°C) Karnovsky solution (2% paraformaldehyde, 100 mM cacodylate buffer pH 7.2, 2.5% glutaraldehyde), mixed by inversion and incubated for 1 h at RT. Fixed cells were harvested (1,000 × g, 2 min, RT), washed three times with cacodylate buffer (100 mM, pH 7.2) (1,500 × g, 5 min, RT) and resuspended in 500 µl of cacodylate buffer. The attachment of cells to poly-L-lysine-coated coverslips was carried out in 24-well plates by centrifugation (1,000 g, 5 min, RT). Then, the samples were washed with 1 ml of cacodylate buffer for 5 min (1,000 g, 5 min, RT). To increase the contrast, the samples were incubated in 2% tannic acid in cacodylate buffer for 1 h at 4°C. Afterwards, the cells were washed again once with 1 ml of cacodylate buffer and three times with H$_2$O for 5 min each (1,000 g, 5 min, RT). The coverslips were divided and transferred into vessels suitable for critical point drying. The samples were dehydrated in a series of ethanol (EtOH) solutions (30%, 50%, 70% and 90% EtOH for 5 min each and six times in 100% EtOH for 5 min), critical-point-dried in CO$_2$, coated with gold palladium and imaged with a JEOL JSM-7500F scanning electron microscope.

## Supporting information

**S1 Table. Label-free quantification of proteins found in the infectome experiments.** Each time point was measured in quadruplicates (01–04).
(XLSX)

**S2 Table. Leishmania protein IDs orthology table.**
(XLSX)

**S3 Table. Dynamicity scores for in the infectome experiments.**
(XLSX)

**S4 Table. Mouse protein clusters for the infectome experiments.**
(XLSX)

 

**S4 Table. GO BP terms overrepresented in the *M. musculus* cluster analysis overlaps.**
(XLSX)

**S6 Table. Hallmark Signature terms overrepresented in the M. musculus cluster analysis.**
(XLSX)

**S7 Table. Leishmania protein clusters for the infectome experiments.**
(XLSX)

**S8 Table. GO BP terms overrepresented in the Leishmania cluster analysis.**
(XLSX)

**S9 Table. Label-free quantification of proteins found enriched in the Leishmania infantum infectome experiment.**
Each time point was measured in quadruplicates (01–04).
(XLSX)

**S10 Table. Label-free quantification of proteins found enriched in the Leishmania major infectome experiment.**
Each time point was measured in quadruplicates (01–04).
(XLSX)

**S11 Table. Label-free quantification of proteins found enriched in the Leishmania mexicana infectome experiment.** Each time point was measured in quadruplicates (01–04).
(XLSX)

**S1 Fig. Infection rate, number of amastigotes per infected macrophage and *Leishmania* per 100 macrophages at different time points of the culture.** In parallel to the *Leishmania spp.*-infected BMDM proteome samples, *Leishmania spp.*-infected BMDM (MOI = 5) were stained with DiffQuick solutions and microscopically analysed to determine infection rates and the number of parasites per infected BMDM at different time points after addition of promastigotes. 8–10 visual fields with 60–120 BMDM each derived from 2 replicates were counted per group. Mean ± SD is shown.
(PDF)

**S2 Fig. *Leishmania spp.* orhtologs . a,** Density plot showing the number of proteins IDs included per OrthoMCL ID for *L. infantum* (green)*, L. major* (pink), and *L. mexicana* (blue). **b,** Barplot of the number of quantified OrthoMCL IDs per infection time course with the color indicating the number of quantified protein IDs associated to the quantified OrthoMCL ID.
(PDF)

**S3 Fig. Relation between proteome variancy and infection time course progression . a,** Heatmap displaying the Spearman correlation coeficient for all measured samples in the *L. infantum* (green), *L. major* (pink) and *L. mexicana* (blue) infection time course experiments. Each row represents the correlation coeficient (light-to-dark blue sale) of each sample across the time course (grayscale). **b,** Violin plots displaying the distribution of the Euclidean distance between each sample from adjacent time points for the *L. infantum* (green), *L. major* (pink) and *L. mexicana* (blue) infection time course experiments. Differences between time courses are tested with Welch's t-test (p-value < 0.05).
(PDF)

**S4 Fig. Protein expression profiles for the three *Leishmania* spp. experiments. a, Heatmap of quantified proteins for the *L. infantum* (green), *L. major* (pink), and *L. mexicana* (blue) experiments.** Each row represents the abundance (z score [LFQ]; yellow-to-blue scale) of each quantified protein across the postinfection timepoints (grayscale). Proteins are annotated as either *M. musculus* (gray) or belonging to *Leishmania* spp. (green, pink, and blue for *L. infantum, L. major,* and *L. mexicana*, respectively)*.* **b,** Line plot of highlighted proteins for the *L. infantum* (green), *L. major* (pink), and

*L. mexicana* (blue) experiments. Each line represents the protein abundance (mean [log2{LFQ}]; y-axis) across the hours postinfection (x-axis). Lines are color coded using, pinkscale, or bluescale for *L. infantum, L. major,* and *L. mexicana* experiments, respectively.
(PDF)

**S5 Fig. *M. musculus* SOM analysis. a,** Barplot showing the number quantified protein IDs before (0: Raw) and after filtering (1: Quantile filter and 2: IQR filter) prior to the SOM analysis for the *L. infantum* (green), *L. major* (pink) and *L. mexicana* (blue) infection time course experiments. **b,** SOM model. Each cell of the 2x3 hexagonal topology grid represents a cluster. Each panel shows, clock-wise, and starting at the upper-left panel: how cells are coloured depending on the overall distance to their nearest neighbors; a pie-chart depicting the representative vectors, where the radius of a wedge corresponds to its magnitude in a particular time point; how cells are coloured depending on the number of protein IDs; each protein ID is depicted in its cell, based on how close they are to the representive vector. **c,** Boxplot showing the distribution of the distance between each protein ID and its winning unit. The mean is shown as a red dot, while the dashed line shows the distance's 75th quantile.
(PDF)

**S6 Fig. *Leishmania Spp.* SOM analysis. a,** Barplot showing the number quantified protein IDs before (0: Raw) and after filtering (1: Quantile filter, 2: IQR filter and 3: Orthology filter) prior to the SOM analysis for the *L. infantum* (green), *L. major* (pink) and *L. mexicana* (blue) infection time course experiments. **b,** SOM model. Each cell of the 2x3 hexagonal topology grid represents a cluster. Each panel shows, clock-wise, and starting at the upper-left panel: how cells are coloured depending on the overall distance to their nearest neighbors; a pie-chart depicting the representative vectors, where the radius of a wedge corresponds to its magnitude in a particular time point; how cells are coloured depending on the number of protein IDs; each protein ID depicted in its cell, based on how close they are to the representive vector. **c,** Boxplot showing the distribution of the distance between each protein ID and its winning unit. The mean is shown as a red dot, while the dashed line shows the distance's 75th quantile.
(PDF)

**S7 Fig. *M. musculus* GO functional analysis. a,** Dot pot showing all significantly overrepresented (FDR<0.05, dark-to-light blue scale) GO terms (y-axis) and their gene ratio (x-axis). The dot size represents the number of included proteins from each particular GO term.
(PDF)

**S8 Fig. *M. musculus* Hallmark signature functional analysis. a,** Dot pot showing all significantly overrepresented (FDR<0.05, blue-to-red scale) Hallmark signature terms (y-axis) and their gene ratio (x-axis). The dot size represents the number of included proteins from each particular term.
(PDF)

**S9 Fig. *Leishmania* GO tBP terns functional analysis.** Dot pot showing all significantly overrepresented (FDR<0.05, blue-to-red scale) GO BP signature terms (y-axis) and their gene ratio per cluster (x-axis) for the *L. infantum* (**a**), *L. major* (**b**) and *L. mexicana* (**c**) experiments. The dot size represents the number of included proteins from each particular term.
(PDF)

**S10 Fig. *In vitro differentiation* of ΔLmx28.2260 and ΔLmx10.0130.** Procyclic promastigote (left panels) and axenic amastigote (right panels) of ΔLmx28.2260 (middle panels) and ΔLmx10.0130 (bottom panels) populations show life cycle stage-specific morphologies, as observed by scanning electron microscopy. LmxCas9 cells (top panels) were analysed as a control.
(PDF)

## Author contributions

**Conceptualization:** Ulrike Schleicher, Falk Butter, Christian J. Janzen.

**Data curation:** Albert Fradera-Sola, Falk Butter.

**Investigation:** Nicolas Hagedorn, Melina Mitnacht, Tobias Gold.

**Resources:** Ulrike Schleicher, Falk Butter, Christian J. Janzen.

**Software:** Albert Fradera-Sola.

**Supervision:** Ulrike Schleicher, Falk Butter, Christian J. Janzen.

**Writing – original draft:** Nicolas Hagedorn, Albert Fradera-Sola.

**Writing – review & editing:** Ulrike Schleicher, Falk Butter, Christian J. Janzen.

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
