## [Decision Letter · Decision Letter 0]

25 Mar 2025

Quantitative proteomics of infected macrophages reveals novel Leishmania virulence factors

PLOS Pathogens

Dear Dr. Janzen,

Thank you for submitting your manuscript to PLOS Pathogens. After careful consideration, we feel that it has merit but does not fully meet PLOS Pathogens's publication criteria as it currently stands. Therefore, we invite you to submit a revised version of the manuscript that addresses the points raised during the review process.

Please submit your revised manuscript within 60 days May 24 2025 11:59PM. If you will need more time than this to complete your revisions, please reply to this message or contact the journal office at plospathogens@plos.org. Please include the following items when submitting your revised manuscript:

We look forward to receiving your revised manuscript.

Kind regards,

Dawn M Wetzel, MD, PhD

Academic Editor

PLOS Pathogens

Margaret Phillips

Section Editor

PLOS Pathogens

Editor-in-Chief

PLOS Pathogens

orcid.org/0000-0003-2946-9497

Editor-in-Chief

PLOS Pathogens

orcid.org/0000-0002-7699-2064

**Additional Editor Comments :**

The reviewers are in good agreement regarding the strengths and weaknesses of the manuscript. They appreciate the effort and detail employed by the authors to study the proteomics of Leishmania infection of macrophages, and feel the findings are significant overall. However, there are issues that prevent further enthusiasm for the manuscript as it currently stands. In particular, the logic and hypothesis regarding why the 3 species were chosen and their infection courses within macrophages are not well-explained. Overall, the manuscript requires a more in-depth, systematic analysis of the described expression and proteomics changes among the 3 Leishmania species used, as indicated by the reviewers. Also, some of the data in the results section requires clarification or supplemental figures to show evidence behind statements that were made. Finally, the discussion should be developed further in accordance with the requests made by the reviewers.

Please let us know if making the requested changes will require extending the deadline for resubmission.

**Journal Requirements:**

At this stage, the following Authors/Authors require contributions: Nicolas Hagedorn, Albert Fradera-Sola, Melina Mitnacht, Tobias Gold, Ulrike Schleicher, Falk Butter, and Christian J. Janzen. Please ensure that the full contributions of each author are acknowledged in the "Add/Edit/Remove Authors" section of our submission form.

https://journals.plos.org/plospathogens/s/submission-guidelines#loc-parts-of-a-submission

4) If your study involves live participants, please insert an Ethics Statement at the beginning of your Methods section, under a subheading 'Ethics Statement'. It must include:

i) The full name(s) of the Institutional Review Board(s) or Ethics Committee(s)

ii) The approval number(s), or a statement that approval was granted by the named board(s).

5) Please upload all main figures as separate Figure files in .tif or .eps format. For more information about how to convert and format your figure files please see our guidelines:

6) We have noticed that you have uploaded Supporting Information files, but you have not included a list of legends. Please add a full list of legends for your Supporting Information files after the references list.

7) Some material included in your submission may be copyrighted. According to PLOSu2019s copyright policy, authors who use figures or other material (e.g., graphics, clipart, maps) from another author or copyright holder must demonstrate or obtain permission to publish this material under the Creative Commons Attribution 4.0 International (CC BY 4.0) License used by PLOS journals. Please closely review the details of PLOSu2019s copyright requirements here: PLOS Licenses and Copyright. If you need to request permissions from a copyright holder, you may use PLOS's Copyright Content Permission form.

Potential Copyright Issues:

i) Figure 1A. Please confirm whether you drew the images / clip-art within the figure panels by hand. If you did not draw the images, please provide (a) a link to the source of the images or icons and their license / terms of use; or (b) written permission from the copyright holder to publish the images or icons under our CC BY 4.0 license. Alternatively, you may replace the images with open source alternatives. See these open source resources you may use to replace images / clip-art:

**Reviewers' Comments:**

Reviewer's Responses to Questions

**Part I - Summary**

Reviewer #1: The authors present a complex proteomic profiling of murine macrophages infected with three different species of Leishmania parasites at different time points. The species were chosen according to the clinical symptoms they cause: L. major causes cutaneous lesions, L. mexicana shows muco-cutaneous involvement and L. infantum presents a visceral tropism. The approach used is pertinent and generates valuable data regarding Leishmania and host cell responses during infection. However, important points are missing or not exploited in a comprehensive manner. In particular, the infection course of each parasite strain used in the macrophages is not described but necessary for the interpretation of the results (see major comments). The methods for data analyses and resulting plots are insufficiently described, and some choices are causing confusion, in particular analyzing host cell and Leishmania data together or separate depending on the figures. The impact of this manuscript could have been significantly increased by an in-depth systems analysis of the identified expression changes (e.g. GO enrichment, STRING network etc), which could render the rather descriptive data biologically more relevant. The discussion is a bit weak: the second paragraph is very repetitive, the third one pushes the idea of species-specific virulence factors, but without actual experimental proof, and the relevance of the approach used - in vitro infections of murine BMDMs - regarding in vivo infection is not discussed.

Reviewer #2: The authors of this paper carry out a comprehensive quantitative mass spectrometric analysis of the proteomes of three Leishmania species and Bone Marrow Derived Macrophages (BMDMs) over the course of an in vitro infection. They select a couple of putative L. mexicana virulence factors for further characterization as a proof of principle for functional significance during the infection. The work, along with publicly searchable database, contributes a valuable resource for the Leishmania, the ‘TriTryp’ community, and all those who study intracellular pathogens. The work is executed with rigor (experiments in quadruplicates) and I appreciated the interrogation of the early time points in the infection (0.5 h and 2 h). While the manuscript is generally written, I believe the authors can do a better job at integrating the results into existing knowledge and better highlighting the new insights gained into the infection dynamics of a Leishmania infection. The sections describing the parasite and macrophage protein expression results are a little anemic.

Reviewer #3: Thank you for the opportunity to review the manuscript entitled "Quantitative proteomics of infected macrophages reveals novel Leishmania virulence factors." Overall, the study presents interesting findings, and I appreciate the effort put into exploring the proteomics of Leishmania virulence factors. However, there are several key areas in the manuscript that require clarification and further development to enhance the overall impact of the work.

**Part II – Major Issues: Key Experiments Required for Acceptance**

Reviewer #1: General comments:

- A proper description of the course of each type of infection must be included. How does each parasite species perform in the macrophages over time, until the time point of 72H but also after? Are they cleared or do they proliferate? Is the dynamic similar for the three species or different? This is essential, as it could explain the different dynamics of expression profiles observed for each parasite species over time. The authors should not only show the number of infected cells (only metrics given in the Fig. 6) at each time point but also the number of parasites per 100 macrophages to better assess the quality of infection. These are key parameters used for in vitro macrophages infection studies.

- The choice of performing the analyses on the host cell and Leishmania proteomes together or separately is not always pertinent in my opinion, and sometimes causes confusion. For example, I don’t see the point of doing the analyses presented in Figure 2 to pool both proteomes, as it renders the heatmaps useless. For the data shown in Figures 3 and 4: were the 9 clusters defined based on the total proteome (macrophage + Leishmania) and then only the mouse or parasite proteins were analyzed, or were the clusters defined on each proteome separately? How was the number of clusters (9 in each case) determined? This information should be clarified and included in the methods section.

Detailed comments:

- Fig 1: Panel C: How were these PCA plots made? Was each infected time point compared to the non-infected control at time point 0, or to non-infected time-points at the same time in culture? This should be clearly explained in the text and methods. The proteome of (uninfected) macrophages in culture is likely to change over time, even when not infected. How was this aspect considered in the analyses?

- I understand that these PCA plots combine the mouse and parasites proteins, but this should be clarified, as it is not obvious from the legend. It would be interesting to have separated PCA plots for only the mouse or the Leishmania proteins (maybe in a supplementary figure).

- Line 125: The authors write “The lower number of L. infantum proteins correlates with its lower invasion rates of macrophages, which resulted in a lower number of parasites in individual BMDMs”. Where is this very important information shown? Do you suggest that it is an explanation for the lower number of L. infantum proteins detected in your approach? How were differences in infectivity considered when assessing the differential proteomes of both host cell and parasite?

- Fig 2: The choice of combining host cell and Leishmania proteomes eludes me. For panel A in particular, mixing the proteomes of the host cells and the pathogen renders the heatmap useless, as we can’t distinguish any pattern/clustering anymore. Panel C, presenting the dynamic of expression of chosen factors over time, is interesting but needs to be put into perspective with the different dynamics of infection, that need to be added. If the result the authors want to illustrate is that some common proteins present different expression dynamics depending on the species, then they should present the dynamic of expression of these factors in the 3 species in the same graph (one graph per protein), at least in supplementary figures.

- Fig 3-4: As mentioned above, it is not clear from the methods section if the SOM clusters were made from the total proteomes (host cell + Leishmania), or only on the host cell for Figure 3 and Leishmania for Figure 4. Given that these figures then show the expression profiles of host cell and pathogen proteomes separately, the SOM analysis should be done separately. Panels A and B should be inverted. As for Fig 2, the biological relevance of the data presented here is unclear. This needs to be rectified by (i) assessing the dynamics of infection of the macrophages with the 3 species, and (ii) including GO term enrichment analysis.

- Line 210: “Overall, we observed different temporal protein expression patterns for the Leishmania spp., with many virulence factors peaking earlier in L. infantum than in L. major and L. mexicana.” The authors need to show if this is due (or not) to differences in infection dynamics between the different parasite species.

- Fig 5: Figure title and legend: “significant expression profile” (title) and “significant protein IDs”. What does ‘significant ‘ mean here? This should be reformulated if this does not refer to statistics. Panel B is not clear: what time points are considered here? The text only mentions “We overlapped all significantly up- and downregulated proteins”. When?

- Fig 6: A method section to describe the process of differentiation from promastigotes to amastigotes in culture and the readout used for successful differentiation (cumulative cell density) is needed. In panels D and H: the percentage of infected cells is not enough to properly follow the infections, as some infected cells could contain very few parasites and other hundreds. The authors need to include the number of parasites per 100 macrophages to assess the distribution of parasites. Panel F: if the dynamic of expression of the protein PFR-2 is used as a read-out for amastigote differentiation in culture, then it should be presented at the beginning of the figure, in a separated panel and for all 3 Leishmania species. Microscopy pictures to illustrate the differentiation of amastigotes (or amastigote-like forms) in culture would be appreciated in Sup. Figures (line 303-306: “The knockout cells were unable to proliferate as axenic amastigotes, although they had the morphological characteristics of amastigotes, such as a rounded cell body and a shortened flagellum”).

- Discussion: line 352-355: “Experiments with this protein also underscore the differences in the infection dynamics among the three species, as this protein was significantly regulated only in L. mexicana but not in L. infantum or L. major”. Here you imply that this protein has no role in the virulence of L. major or L. infantum, but the only way to know it is a loss-of-function study in L. major and L. infantum. Idem at 366: “Given the differences in the protein dynamics of LmxM.28.2260 and its orthologs in L. major and L. infantum, it is possible that the protein plays species-specific roles in the infection process”. Either show experimental evidence for this or strongly tone down.

Reviewer #2: Fig. 2 panel -This panel not pretty but informative

Fig. 3 – I am not sure I agree with the interpretation of the self-organizing map analysis and with the assertion that that ‘infection response of the host cell to different parasite species is generally similar’. One sees similar trends (as expected you lump the three datasets in the same clustering analysis), but most protein IDs are not shared. The more striking observation is that the proteins underlying those trends are mostly specific to the infections by different species of parasites. The response seems to be quite heterogeneous and infection species-specific! Only 383 mouse proteins behave identically across all 9 clusters. An attempt to understand/dissect the apparent species-specific response would be important here.

ln 194 – In the same vein, the statement “ there is a large overlap in M. musculus proteome expression changes, with a similar temporal pattern” reflects an anticipated result given the nature of the analysis which is aimed at doing exactly that.

Fig. 3A – I appreciate what the authors are trying to represent in Fig. 3A. but the bar plot (stacked or not) is not the most accurate given the redundant proteins identified in the response to species-specific infections. I believe the Venn diagrams I panel C are sufficient.

The same comment applies to Fig. 4A (despite the fact that the overlaps are less significant here)

Ln 210-212 ‘Overall, we observed different temporal protein expression patterns for the Leishmania spp., with many virulence factors peaking earlier…’. This statement is made as a summary for a section where no reference to virulence factors is made.

What is the definition of virulence factors in this paper? And consequently, what criteria were used for selecting the proof of principle proteins? Such filters, and the resulting subsets, would be of interest to the readers.

Reviewer #3: INTRODUCTION

• Line 48: The statement made requires citation, as there is still ongoing debate regarding the relationship between species and pathology in Leishmania. A reference would strengthen this assertion.

• Line 60: The statement is misleading as it implies that only Lutzomyia and Phebotomus species are involved in transmission. Please revise to reflect the involvement of other genera in Leishmania transmission.

• Line 80: While I understand that this study focuses on L. major, L. mexicana, and L. infantum, the manuscript would benefit from a broader discussion of other epidemiologically relevant species. Briefly contextualizing this research within a broader species framework would provide greater relevance to the field.

• Line 90: The hypothesis for using only these three species is not clearly defined. It would be helpful to explain the rationale for selecting these species in particular. The readers should also gain better insight into the utility of quantitative proteomics in the context of Leishmania research, as this is not well conveyed in the introduction.

METHODS

• Line 384: Species names should be italicized for consistency with scientific writing standards. Also, the WHO nomenclature for the strains used is missing and should be provided for clarity.

• Line 404: The rationale behind the selection of the multiplicity of infection (MOI) is not sufficiently explained. Was the same MOI used for all three species? If so, why? The assumption of no differences in species infectivity is not supported, and this needs to be addressed more explicitly.

• Line 409: The methodology section lacks clarity regarding the timing of proteome extraction. It would be helpful to include information on the time points of infection used, the number of replicates, and how infection was quantified and selected. Providing infection plots would be beneficial here.

• Line 470: The rationale for transfecting L. mexicana is not well-justified at this point in the manuscript. While this is clarified in the results, it would be helpful for the reader to understand the reason behind this decision earlier in the methods section.

RESULTS

• The results are intriguing, particularly the identification of shortlisted virulence factors. However, validation of these findings using additional methods is crucial. The data would benefit from further validation via independent approaches, especially considering the power of the proteomics data at hand.

• Given the comprehensive nature of the proteomics data, I would expect a more thorough exploration of the protein interactome. The inclusion of detailed analyses using well-established tools, such as AlphaFold Multimer, HADDOCK, ClusPro, and PyMOL for visualization, would elevate the findings significantly. Furthermore, using RVFScan for virulence factor gene detection and incorporating BioNumerics or Python for data analysis could provide additional insights into protein interactions and host-pathogen dynamics.

DISCUSSION

• The discussion section feels underdeveloped and overly speculative. While the authors provide some interpretation of the results, I would appreciate a deeper analysis of how these findings might translate from mice models to human infection. Additionally, the authors should be more transparent in discussing the limitations of their study. Given the scope of the research, there are several limitations that are not adequately addressed.

**Part III – Minor Issues: Editorial and Data Presentation Modifications**

Reviewer #1: - Line 33: “…demonstrating the utility of our dataset”: this formulation is unclear and should be rephrased.

- Line 64-66: “Once inside the host cell, the parasites interfere with the phagocytic pathway, resulting in a delay in phagolysosome maturation and, thus, elimination of the parasites”: the sentence is unclear, rephrase.

- Line 94-96: “Only one study has investigated the dynamic changes that occur during the process of infection with L. amazonensis and L. major parasites in human macrophages, via transcriptomics”: I would reformulate to “only one study has used transcriptomics to investigate…”

- Line 100: “murine BMDMs”

- Line 132: “The lower number of quantified proteins among the three species also coincided with a lower number of species-exclusive proteins”: this sentence is unclear, rephrase.

- Line 152: “Gini score” (not Gine)

- Line 251: “We tagged the gene in a cell line expressing GFP-GAPDH to investigate the localization of our candidate in glycosomal compartments of L. mexicana. Our microscopy images revealed that the LmxM.28.2260 signal clearly overlapped with that of GAPDH”. This formulation is unclear, as we don’t understand that GAPDH is used as a glycosomal marker.

- Line 257: “altered” should be replaced by modified or changed

- Line 295: “Although L. infantum and L. major orthologs were also quantified, we only measured a significant difference between time points in L. mexicana”. This is unclear: you didn’t detect the protein in the other species, or its expression wasn’t changing over time?

- Line 327-342: the paragraph is looping, repeating the same idea in 3 different formulations (the pathology is driven by the parasite not the host). Also, it should be discussed in this paragraph that this is an in vitro infection system and its shortcoming. How relevant are your data for an in vivo setting?

- Line 409: Protein extracts (not proteome)

- Line 453: “For protein dynamicity analysis, protein abundance stability across embryonic developmental stages was evaluated via the Gini coefficient”: embryonic developmental stages?? I guess this comes from a previous publication. To be removed.

Reviewer #2: In the discussion of Fig.2 results, abundance and intensity are used interchangeably. Perhaps stick to one?

Reviewer #3: See above

PLOS authors have the option to publish the peer review history of their article (what does this mean? ). If published, this will include your full peer review and any attached files.

**Do you want your identity to be public for this peer review?** For information about this choice, including consent withdrawal, please see our Privacy Policy .

Reviewer #1: **Yes:** Isabelle Louradour

Reviewer #2: No

Reviewer #3: No

**Figure resubmission:**

**Reproducibility:**



---

## [Decision Letter · Decision Letter 1]

18 Nov 2025

PPATHOGENS-D-25-00202R1

Quantitative proteomics of infected macrophages reveals novel Leishmania virulence factors

PLOS Pathogens

Dear Dr. Janzen,

Thank you for submitting your manuscript to PLOS Pathogens. After careful consideration, we feel that it has merit but does not fully meet PLOS Pathogens's publication criteria as it currently stands. Therefore, we invite you to submit a revised version of the manuscript that addresses the points raised during the review process.

We look forward to receiving your revised manuscript.

Kind regards,

Dawn M Wetzel, MD, PhD

Academic Editor

PLOS Pathogens

Margaret Phillips

Section Editor

PLOS Pathogens

Sumita Bhaduri-McIntosh

Editor-in-Chief

PLOS Pathogens

orcid.org/0000-0003-2946-9497

Michael Malim

Editor-in-Chief

PLOS Pathogens

orcid.org/0000-0002-7699-2064

**Additional Editor Comments:**

Please make the minor revisions requested by Reviewer 1.

Reviewer 2 is unavailable to re-review the manuscript, but the editor feels that their comments have been adequately addressed.

**Journal Requirements:**

Please ensure that the Title in your manuscript file and the Title provided in your online submission form are the same.

**Reviewers' Comments:**

Reviewer's Responses to Questions

**Part I - Summary**

Reviewer #1: This study presents a complex proteomic profiling of murine BMDMs infected with three different species of Leishmania parasites, chosen because they induce different clinical manifestations, at different time points. The work presented is well executed and provides a valuable resource for researchers investigating Leishmania parasites. In the revised version of their manuscript, the authors added important informations and analyses (in particular a better description of the dynamic of infection of each parasite species in the BMDMs) and modified the text accordingly. I do appreciate that some previous statements were cleraly toned down, to avoid over-interpretation of data.The resulting manuscript is more accurate and to my opinion only requires minor modifications.

Reviewer #4: (No Response)

**Part II – Major Issues: Key Experiments Required for Acceptance**

Reviewer #1: (No Response)

Reviewer #4: (No Response)

**Part III – Minor Issues: Editorial and Data Presentation Modifications**

Reviewer #1: - I thank the authors for adding the Supp. Figure 1, showing the dynamic of infection of each parasite species in BMDMs other time, answering my biggest concern. These results actually show that infantum tends to be cleared other time in this cells, contrary to mexicana who seems to establish a productive infection and proliferate, and could actually explain, at least in part, the dynamic of expression of some proteins (there is just less parasites for infantum, not really less protein per parasite). This has been sort of noted in the manuscript, but could be better integrated in relation to some results (in particular Fig 6). Of note, a graph showing the number of parasites per 100 macrophages (which can be deduced from the percentage of infected cells and the mean number of parasites per infected cell) would be appreciated (and was asked previously as a major comment).

- In relation to that point, I suggest to change this sentence to reflect this result: line 294: “This finding was consistent with our previous observation of species-specific protein expression dynamics during the time course of infection (Fig.4) and also correlates with the dynamic of infection profiles of the 3 parasites (Supp. Fig1).

- As it is written currently in the method section, the SOM analysis is still not super clear to me: I suggest that the authors simply include the answer they gave to the reviewing inside the method section: “For the self-organizing map (SOM) analysis (Figures 3 and 4), host and parasite datasets were analysed independently. The number of clusters (nine per species) was determined empirically, based on both intracluster distances (i.e., similarity of protein expression profiles within clusters) and neighbourhood distances between clusters to ensure meaningful separation”

- Concerning the Fig 5 title and legend: “significant expression profile” (title) and “significant protein IDs”. I thank the authors for their explanation of what is significant in their study, but the point I was making here was purely semantic: there is no such thing as a “significant ID” or “significant profile”. I saw that this was modified in the text, but please modify the title of the figure (I suggest “Pairwise differential… with significantly modulated expression profiles”

- For the GFP-GAPDH sentence: even if I agree with the authors that GAPDH is a classical marker of glycosomes in trypanosomatids, it will not be obvious to every reader, especially those not familiar with the model. I think you just need a clear sentence in that part of the text saying that GFP-GAPDH is used as a marker of the glycosomal compartment. Your sentence “We tagged the gene in a cell line expressing GFP-GAPDH to investigate the localization of our candidate in glycosomal compartments of L. mexicana.” is ambiguous.

Reviewer #4: (No Response)

PLOS authors have the option to publish the peer review history of their article (what does this mean? ). If published, this will include your full peer review and any attached files.

**Do you want your identity to be public for this peer review?** For information about this choice, including consent withdrawal, please see our Privacy Policy .

Reviewer #1: **Yes:** Isabelle Louradour

Reviewer #4: No

**Figure resubmission:**
---

## [Editor Report · Decision Letter 2]

26 Jan 2026

Dear Dr Janzen,

We are pleased to inform you that your manuscript 'Quantitative proteomics of infected macrophages reveals novel Leishmania virulence factors' has been provisionally accepted for publication in PLOS Pathogens.

Best regards,

Dawn M Wetzel, MD, PhD

Academic Editor

PLOS Pathogens

Margaret Phillips

Section Editor

PLOS Pathogens

Sumita Bhaduri-McIntosh

Editor-in-Chief

PLOS Pathogens

orcid.org/0000-0003-2946-9497

Michael Malim

Editor-in-Chief

PLOS Pathogens

orcid.org/0000-0002-7699-2064
---

## [Editor Report · Acceptance letter]

Dear Dr Janzen,

We are delighted to inform you that your manuscript, "Quantitative proteomics of infected macrophages reveals novel Leishmania virulence factors," has been formally accepted for publication in PLOS Pathogens.

Best regards,

Sumita Bhaduri-McIntosh

Editor-in-Chief

PLOS Pathogens

orcid.org/0000-0003-2946-9497

Michael Malim

Editor-in-Chief

PLOS Pathogens

orcid.org/0000-0002-7699-2064